# Anti-Inflammatory and Antioxidant Properties of Carvacrol and Magnolol, in Periodontal Disease and Diabetes Mellitus

**DOI:** 10.3390/molecules26226899

**Published:** 2021-11-16

**Authors:** Georgiana Ioana Potra Cicalău, Petru Aurel Babes, Horia Calniceanu, Adelina Popa, Gabriela Ciavoi, Gilda Mihaela Iova, Mariana Ganea, Ioana Scrobotă

**Affiliations:** 1Doctoral School of Biomedical Science, University of Oradea, 1st University Street, 410087 Oradea, Romania; cicalau.georgiana@gmail.com; 2Department of Dental Medicine, Faculty of Medicine and Pharmacy, University of Oradea, 1st Decembrie Street, 410073 Oradea, Romania; gciavoi@uoradea.ro (G.C.); gilda_iova@yahoo.ro (G.M.I.); ioana_scrobota@yahoo.com (I.S.); 3Department of Periodontology, Faculty of Dental Medicine, Victor Babes University of Medicine and Pharmacy, 300041 Timisoara, Romania; 4Periodontal and Periimplant Diseases Research Center “Prof. Dr. Anton Sculean”, Faculty of Dental Medicine, Victor Babes University of Medicine and Pharmacy, 300041 Timisoara, Romania; 5Department of Orthodontics, Victor Babes University of Medicine and Pharmacy, 300041 Timisoara, Romania; 6Orthodontic Research Center (ORTHO-CENTER), Faculty of Dental Medicine, Victor Babes University of Medicine and Pharmacy, 300041 Timisoara, Romania; 7Department of Pharmacy, Faculty of Medicine and Pharmacy, University of Oradea, 1st Decembrie Street, 410073 Oradea, Romania; madafarm2005@yahoo.com

**Keywords:** carvacrol, magnolol, periodontitis, diabetes, anti-inflammatory, antioxidant, antimicrobial, anti-osteoclastic, anti-diabetic, toxicity

## Abstract

Periodontal disease and diabetes mellitus are two pathologies that are extremely widespread worldwide and share the feature of chronic inflammation. Carvacrol is a phenolic monoterpenoid, produced by a variety of herbs, the most well-known of which is *Origanum vulgare*. Magnolol is a traditional polyphenolic compound isolated from the stem bark of *Magnolia officinalis*, mainly used in Chinese medicine. The purpose of this paper is to review the therapeutic properties of these bioactive compounds, in the treatment of periodontitis and diabetes. Based on our search strategy we conducted a literature search in the PubMed and Google Scholar databases to identify studies. A total of one hundred eighty-four papers were included in the current review. The results show that carvacrol and magnolol have anti-inflammatory, antioxidant, antimicrobial, anti-osteoclastic, and anti-diabetic properties that benefit both pathologies. Knowledge of the multiple activities of carvacrol and magnolol can assist with the development of new treatment strategies, and the design of clinical animal and human trials will maximize the potential benefits of these extracts in subjects suffering from periodontitis or diabetes.

## 1. Introduction

Periodontal disease is the subject of a public health problem, being a pathology that joins general conditions. This pathology, also called periodontitis, is an infectious disease of the oral cavity, characterized by irreversible destruction of the tooth-supporting structures: Alveolar bone, periodontal ligament, and cementum [1]. Periodontitis is one of the major causes of tooth loss, which endangers the functions of the stomatognathic system: Mastication, phonation, physiognomy, aesthetics, as well as self-confidence, self-esteem, and quality of life of patients [2,3]. According to a 2016 survey fulfilled by the Global Burden of Disease (GBD), severe periodontal disease was the 11th most widespread disease on the globe [4]. In the case of periodontitis, a prevalence of 20–50% worldwide has been noted [5].

Diabetes is characterized by the body′s inability to control blood glucose levels. Type 1 and type 2 diabetes are the forms of diabetes that have been stated to act upon periodontium. In type 1 diabetes, pancreatic β-cells do not synthesize insulin or synthesize an insufficient amount, patients are treated with insulin, and therefore this type of diabetes is also known as insulin-dependent diabetes mellitus (IDDM). Type 2 diabetes is also called non-insulin-dependent diabetes (NIDDM) and is characterized by a deficiency of insulin receptors [6]. The prevalence of diabetes is constantly rising on all continents and can be classified as an epidemic, due to alarming proportions: 415 million people have been diagnosed with diabetes and it is expected that the values will increase to about 640 million in 2040 [7]. Diabetes mellitus will be the seventh leading cause of mortality by 2030, as per World Health Organization (WHO) statistics [8].

Herbal extracts have been considered therapeutic elements since ancient times. Currently, more and more natural compounds, essential oils, and vegetable extracts have attracted interest from researchers, due to their antioxidant properties and benefits for the wellbeing of the human body [9].

Carvacrol is a phenolic monoterpenoid, produced by many herbs, the best known of which are *Origanum vulgare* (Greek oregano, wild marjoram), *Origanum majorana* (marjoram), *Satureja hortensis* (summer thyme), *Thymus vulgaris* (thyme), and *Satureja montana* (winter thyme) [10]. Carvacrol has long been recognized as a component of oregano essential oil, being one of its most investigated components [11]. Thymol is a structural isomer of carvacrol, with the hydroxyl group (–OH) in the second position and similar characteristics to those of carvacrol [12]. Carvacrol possesses anti-inflammatory, antioxidant, and antibacterial properties [13,14]. At the same time, carvacrol has other biological properties, being anti-diabetic, antifungal [15], antitumor [16], antimutagenic [17], analgesic [18], anti-hepatotoxic [19], cardioprotective [15], and antiparasitic [20].

Magnolol is a binaphthalene polyphenolic compound, isolated from the stem bark of *Magnolia officinalis*, being a traditional extract, used mainly in Chinese medicine. Honokiol is the structural isomer of magnolol [21]. Magnolol was first isolated in 1930 from the magnolia root by Sugii, a Japanese scientist, and was first synthesized by Holger Erdtman and Johan Ludvig Runeberg, two Swedish scientists, using p-allylphenol as a raw material. *Magnolia officinalis* has been described since ancient times, in the *“Shennong Herbal Classic”*, dating from Qin and Han Dynasty, around 221 B.C. to 220 A.D. Magnolol is widely used in Oriental medicine [19]. There is ample evidence that magnolol exerts a wide variety of beneficial pharmacological activities. *Magnolia officinalis* extract has plentiful attributes, such as anti-inflammatory [22,23,24], antioxidant [25], antibacterial [26], anti-osteoclastic [23], antianxiety [27], anti-diabetic [28], antiplatelet, and anticarcinogenetic [29]. Figure 1 shows the chemical structure of carvacrol [30] and magnolol [31].

The purpose of this manuscript is to conduct a literature review on the therapeutic properties of two natural bioactive extracts, carvacrol and magnolol, as well as their impact on periodontal disease and diabetes mellitus. The aim of this assessment is to evaluate the pathogenesis of periodontitis and diabetes, as well as the interdependence between periodontitis and diabetes, and the biological effects of these extracts in these diseases. Last but not least, we will investigate the toxicity of carvacrol and magnolol at various doses. The objective of this review, in summary, is to create a synthesis of the effects of carvacrol and magnolol on periodontitis and diabetes, based on literature research.

## 2. Materials and Methods

### 2.1. Search Strategy

We conducted a review of the properties of bioactive plant extracts, carvacrol and magnolol, against periodontal disease and diabetes mellitus.

This narrative review was conducted using two databases: PubMed and Google Scholar. In order to identify the most revealing articles, the search strategy included combined keywords such as “carvacrol” or “magnolol”, “anti-inflamatory” or “antioxidant” or “antimicrobial” or “anti-osteoclastic” or “bone resorption” or “anti-diabetic”, “periodontitis” or “gingivitis”, “diabetes” or “hyperglycemia”. The databases were searched for studies published between 1986 and 2021.

The search strategy was designed to identify in vivo or in vitro studies on the anti-inflammatory, antioxidant, antimicrobial, anti-osteoclastic, anti-diabetic, and toxicity effects of carvacrol and magnolol on the treatment of various inflammatory diseases, particularly periodontal disease and diabetes mellitus.

### 2.2. Study Selection and Eligibility Criteria

The authors carefully reviewed the articles found and selected the ones that were the most informative in terms of the topic they were looking for. Although there was no restriction on the year of publication for the studies included, the majority of the papers were published after 2010.

All electronically searched titles, selected abstracts and full-text publications were independently reviewed by a minimum of two reviewers. We included in our manuscript papers containing the keywords mentioned. The inclusion criteria were in vivo or in vitro studies, the use of carvacrol or magnolol, the treatment of periodontitis or diabetes or other inflammatory diseases, and the evaluation of anti-inflammatory, antioxidant, antimicrobial, anti-osteoclastic, anti-diabetic action, or toxicity of these extracts. The selected in vivo studies have been conducted on both animals and humans. Exclusion criteria were studies that did not meet the required criteria. Furthermore, we excluded papers that were published in languages other than English. Disagreements over whether texts fit the inclusion or exclusion criteria were resolved through consensus.

Through database research, 438 records were identified. After removing 151 duplicates, as well as 86 irrelevant articles whose theme did not match the standards of this review, we proceeded to read the remaining 201 titles and abstracts eligible for inclusion. We removed 17 articles not meeting all the inclusion criteria. Ultimately, based on our search strategy, we included 187 references that met the criteria of the study. One hundred and sixty-five of these are found in the PubMed database and 14 in the Google Scholar database. The remaining number to complete the total is represented by 7 sites and 1 book. A flow chart regarding the selection criteria of the articles that were taken into account for this review is illustrated in Figure 2.

A section on bibliometric information was included to increase the value of the research. The articles’ quality was evaluated based on the number of citations in the literature and the impact factor of the journals where they were published. As shown in Figure 1, the percentage distribution of the articles included in the study was calculated based on their citations number. Thus, 46% have been quoted 1–100 times, 25% have been quoted 100–200 times, 16% have been quoted 200–500 times, 8% have between 500 and 1000 citations, and 5% have over 1000 citations.

Simultaneously, the articles were classified based on *the* impact factor of the journals where they were published. Thus, 92 articles had an impact factor of 3.00 to 6.00, 43 publications had an impact factor of 1.00 to 3.00, 35 had an impact factor of 6.00 to 9.00, and 17 had an impact factor greater than 9.00. Figure 2 depicts the distribution of articles based on the journals’ impact factor.

## 3. Pathogenesis of Periodontal Disease and Diabetes Mellitus

Periodontal disease is an inflammatory condition of the superficial or deep marginal periodontium, with the primary etiological factor of bacterial biofilm. Periodontal pathogens, represented mainly by the anaerobic flora, induce cellular inflammation, local edema, and consequent vasodilation through their products [32]. Chronic inflammation from gingivitis or periodontitis degrades the supporting structures of the tooth, also acting on the alveolar bone, and in severe cases reaching the avulsion of the teeth [33,34].

Periodontal disease is a pathology with a very high prevalence among the population and is a non-specific inflammatory condition. It is a condition closely related to other systemic inflammatory or non-inflammatory disorders, including coronary heart disease and diabetes [35].

Periodontal pathogens can be grouped into bacterial complexes according to their properties and pathogenicity. One of the oldest classifications of subgingival bacteria was made by Socransky, who grouped periodontopathogenic bacteria into complexes, associated with different periodontal statuses. This classification is still valid today and can be observed in Figure 3 [36].

Those bacteria that rapidly colonize the bacterial plaque, with the ability to adhere to the film, are bacteria from the green, purple, or yellow complex. The orange complex comprises moderately pathogenic periodontal microorganisms such as *Fusobacterium nucleatum* (*F. nucleatum*) and normally occurs after the first colonizers appear in the bacterial plaque [37]. They associate with other bacteria to colonize the gingival sulcus. The bacteria in the red complex have the greatest pathogenicity: *Porphyromonas gingivalis* (*P. gingivalis*), *Treponema denticola* (*T. denticola*), and *Tannerella forsythia* (*T. forsythia*), the most important in the periodontal disease of adults [38]. The pathogenicity of these bacteria increases significantly through the production of various enzymes and toxins. The loss of gingival attachment and the increase in the depth of the periodontal pockets is due to bacteria belonging to the orange complex. Through their metabolism, they provide living conditions for microorganisms in the red complex, strictly anaerobic bacteria, which multiply in the crevicular groove. The presence of bacteria in the red complex and the identification of *Aggregatibacter actinomycetemcomitans* (*A. actinomycetemcomitans*) is evidence of the final colonization of the periodontium [39].

The pathogenesis of periodontal disease would therefore be the result of dysbiosis, caused by ecological stress, generated by the multiplication of numerous periodontal pathogens [40].

In the etiopathogenesis of periodontal disease, complex interactions are involved between numerous mediators of inflammation and mediators of tissue remodeling [41]. Lipopolysaccharides (LPS) are the main constituent of the outer shell of Gram-negative microorganisms, playing a role in the production of many cytokines, such as tumor necrosis factor alpha (TNF-α), interleukin-1β (IL-1β), and interleukin-6 (IL-6) [42,43]. These cytokines infiltrate the gingival tissue and cause local inflammation. Matrix metalloproteinases (MMPs) are a group of host factors, involved in periodontal pathology, being incriminated for the degradation of collagen and the extracellular matrix in periodontal tissue [44]. In addition, gelatinases such as gelatinase A (MMP-2) and gelatinase B (MMP-9) have been associated with periodontitis [45].

During the inflammation generated by periodontitis, there is a significant increase in interleukin expression (IL-1β, IL-6) and transforming growth factor-1β (TGF-1β), which play a key role [46]. At the same time, an alarming increase in growth factors has been observed in diabetes [47,48]. IL-6 is produced by most cells of the immune system and can have anti-inflammatory or pro-inflammatory effects, depending on the circumstances in which it is secreted. The anti-inflammatory effect of IL-6 is mediated by the inhibition of TNF-α and interleukin-1 (IL-1), but also by the activation of agonist receptors of IL-1 and interleukin-10 (IL-10). On the other hand, IL-6 is the mediator of the induction of acute phase proteins, such as the C-reactive protein (CRP). IL-6 is responsible, among other things, for the differentiation and proliferation of B lymphocytes, for the differentiation of monocytes in macrophages, and for the induction of osteoclast formation [49,50]. The most important function of interleukin (IL) is the regulation of bone metabolism [51].

Last but not least, TGF-1β is a multifunctional cytokine with a pleiotropic effect, similar to IL-6, both pro-inflammatory and anti-inflammatory. It stimulates inflammation by chemotactism for monocytes, neutrophils, or lymphocytes and stimulates the production of inflammatory cytokines (IL-1, IL-6) [52,53]. As an anti-inflammatory cytokine, it plays a role in suppressing the humoral response. TGF-1β is secreted by lymphocytes, monocytes, neutrophils, and platelets, making it a very important molecule in wound healing and tissue regeneration [54].

A lack of insulin causes diabetes. Insulin deficiency is generated by insufficient insulin production in IDDM or by insulin resistance in NIDDM, leading to hyperglycemia. Elevated blood sugar levels are associated with disruption of carbohydrate metabolism, which is controlled by enzymes [55].

Chronic hyperglycemia and hyperlipidemia activate oxidative stress, which causes diabetes, cardiovascular, renal, or ocular complications [56]. The liver is the organ that accounts for glucose utilization (30–60% of glucose intake) and regulates blood glucose levels. Glucose homeostasis is maintained through carbohydrate metabolism pathways, such as aerobic oxidation, anaerobic glycolysis, and glycogen synthesis [57].

Complications of diabetes include cardiovascular disease, neuropathy, diabetic nephropathy, retinopathy, and diabetic foot gangrene [58]. In the oral cavity, there are characteristic manifestations of diabetes: Halitosis, xerostomia, sialadenitis, cheilitis, glossodynia, ulcers, increased incidence of infections, and delayed wound healing [59].

The relationship between oral infections and diabetes is far from being fully elucidated by the medical community. However, there are theories that chronic hyperglycemia and hypersecretion of prostaglandins E2 (PGE2) and TNF-α are due to the accumulation of advanced glycation end products (AGEs) [60,61]. A change in collagen metabolism was also observed because of increased collagenase activity and decreased collagen synthesis [62].

Both pathologies have chronic inflammation as a common feature [63,64]. Several studies have shown the bidirectional relationship between periodontal disease and diabetes mellitus [65,66,67]. Both pathologies are extremely widespread worldwide, but the mechanisms that link them are not fully understood [68].

Diabetes is a clinical syndrome characterized by hyperglycemia, which affects all age groups. NIDDM significantly increases the risk of developing periodontal disease and has been suggested to modulate oral microbial communities. The increase in bacterial load could explain the risk of periodontal disease in diabetics [69]. Other studies have shown that NIDDM alters the subgingival bacterial community through inflammation and high blood sugar levels in the crevicular fluid [70]. This would explain the changes in the sulcus in the case of diabetes, with the crevicular groove being a reservoir of bacterial growth [71,72,73,74].

An observational research study observed that periodontitis had a higher predominance in patients suffering from diabetes, compared to patients without hyperglycemia, regardless of differences in age or sex [75]. There is clear evidence that diabetes is a risk factor for gingivitis and periodontitis, and high blood sugar levels are a determining factor in this two-way relationship [6]. Increased values of inflammation have been reported in patients with poor diabetes control [76]. Diabetes is considered to be the only systemic condition with a positive association in terms of loss of gingival attachment [77]. The hyperglycemic environment causes the thickening of the basement membrane of the capillaries, the alteration of the oxygen distribution in the tissues, and the elimination of toxic products. Neutrophil hypofunction alters host defense mechanisms and discredits the immune system against infections [78].

Hyperglycemia modifies fibroblast metabolism, inhibits osteoblastic cell proliferation, and compromises bone healing. Elevated blood glucose levels are a vector for the production and accumulation of AGEs. AGEs bind to monocytes and macrophages, causing them to release several pro-inflammatory cytokines such as IL-1β, TNF-α, and PGE2, causing tissue damage [48,79]. Therefore, diabetes contributes to the aggravation of periodontal pathology through increased levels of inflammation, oxidative stress, and changes in the body’s defenses.

In the case of periodontal disease, several inflammatory molecules are released, such as IL-1β, IL-6, interleukin-8 (IL-8), LPS, TNF-α, and PGE2. These molecules are able to interplay with free fatty acids, lipids, and AGEs, all characteristic of hyperglycemia. Thus, some intracellular pathways associated with insulin resistance are affected, such as nuclear factor-kappa B (NF-κB), the inhibitor of kappa B kinase (IκB), or the inhibitor of kappa B kinase β (IκBβ) [80].

Systemic levels of inflammation mediators, such as CRP, TNF-α, and IL-6, are elevated in periodontal disease and may represent the link between diabetes and periodontitis [74,81,82,83,84]. Another association between periodontitis and hyperglycemia is oxidative stress, which can activate pro-inflammatory pathways similar to both diseases. [85]

Patients with diabetes have an increased risk of developing periodontal disease, especially when they do not control their blood sugar levels. Prevention and treatment of pathologies in the superficial and deep marginal periodontium must be judiciously considered in the management of patients with hyperglycemia. It has been shown that keeping blood sugar levels under control has advantages over periodontal disease, while the treatment of periodontal disease improves metabolism in patients with diabetes [86]. Clear health strategies need to focus on periodontal disease, as a risk factor for diabetes and cardiovascular disease, through prevention and treatment programs for all chronic infections [87].

## 4. Chemistry of Carvacrol and Magnolol

Carvacrol is a liquid phenolic monoterpenoid, present in the essential oil of oregano (*Origanum vulgare*), thyme (*Thymus vulgaris*), pepper (*Lepidium flavum*), wild bergamot (*Citrus aurantium var. Bergamia Loisel*), and other plants [88]. It is also known as 5-Isopropyl-2-methylphenol [30]. Other synonyms for carvacrol are isopropyl-o-cresol, p-cymen-2-ol, 2-hydroxy-p-cymene, 5-isopropyl-2-methylphenol, or iso-thymol [35]. Carvacrol is registered in the IUPAC (International Union of Pure and Applied Chemistry) chemical nomenclature of organic compounds under the name 2-methyl-5-propan-2-ylphenol, with the molecular formula C_10_H_14_O and the CAS identification number 499-75-2 [89]. It has a molecular weight of 150.22 g/mol, a density of 0.976 g/mL at 20 °C, and a boiling point of 236–237 °C [30]. It is insoluble in water, but very soluble in ethanol, acetone, and diethyl ether. Commercial carvacrol is synthesized by chemical and biotechnological methods [90].

Magnolol is a bioactive plant extract, isolated from the bark and root of various species of magnolia, among which we mention *Magnolia officinalis* [30]. This substance of natural origin is a binaftelic polyphenolic compound, also known as 2,2′-Bichavicol or 5,5′-Diallyl-2,2′-biphenyldiol [91]. Magnol is registered in the chemical nomenclature IUPAC of organic compounds under the name 2-(2-hydroxy-5-prop-2-enylphenyl)-4-prop-2-enylphenol, with the molecular formula C_18_H_18_O_2_ and CAS identification number 528-43-8. It has a molecular weight of 266.33 g/mol, water solubility of 1.24 mg/L at 25 °C, and a boiling point of 101.5–102 °C [92].

## 5. Biological Activities of Carvacrol and Magnolol on Periodontitis and Diabetes

Over time, a wide variety of therapeutic methods for the removal of periodontal disease have emerged. One of the most common approaches is mechanical treatment and periodontal surgery, in order to annihilate the microbial load on the periodontium. However, this approach is not always optimal, as periodontal disease is immunogenetically modulated and therefore requires adjuvant therapies [93]. The increased incidence of marginal periodontitis, increased resistance of Gram-negative bacteria to routine antibiotics, and even their side effects motivate researchers to discover new treatment schemes for the prevention and treatment for this illness [94].

Hence the emergence of new herbal medicine formulas, with bioactive molecules, would be beneficial for minimally invasive treatment, simple and predictable, but also with prophylactic potential in the occurrence of marginal periodontitis. Natural medicines consist of plant extracts that are considered to have therapeutic properties. At present, phytotherapy is gaining more and more followers, due to the complex action of the extracts, minimal side effects, and low cost compared to synthetic drugs. At the same time, modern medicines can generate resistance to antibiotics, so herbal treatments are an alternative in combating various diseases of the body and oral cavity [95].

Plant extracts have been paid more and more attention on account of their anti-inflammatory and antibacterial properties and their role in modulating the inflammatory response. Recent research also shows that certain flavonoids have particularly beneficial properties [77]. In recent years, more and more plant extracts have been scientifically investigated in terms of their effect on the bacterial flora of periodontal disease. Many of these studies are experimental research on rats, as this animal model has similar histological, immunological, and biochemical mechanisms to those found in humans [96,97,98,99].

A number of natural extracts have been shown to improve the symptoms of diabetes and chronic marginal periodontitis. Two of these extracts are carvacrol and magnolol [100,101]. An in vivo study showed that carvacrol improves experimentally induced periodontitis in rats and analyzed the effect of intragastric (IG) administration of carvacrol on alveolar bone resorption, using radiographic examinations. The use of carvacrol in small doses is safe and helpful in the treatment of periodontal disease. The results showed that carvacrol protects gingival tissue in rats with periodontal disease, which is mediated by carvacrol through the inhibitory effect on inflammation and degradation of periodontal tissue. Carvacrol also reduces the inflammatory reaction and expression of MMP-9 [102]. Other studies have used carvacrol incorporated into herbal periodontal gels to treat experimentally induced periodontitis in rats and it has been shown that local application of carvacrol has reduced alveolar bone resorption [103,104].

Magnolol has been shown to reduce hyperglycemia and alleviate the complications of diabetes [105]. It has also been shown to relieve the accumulation of stress thanks to its antioxidant properties [90] and reduce inflammation in ligature-induced marginal periodontitis in rats [88]. Last but not least, magnolol reduces the inflammation induced by P. gingivalis LPS in macrophages [106]. Figure 4 summarizes the properties of carvacrol [13,14,15,16,17,18,19,20] and magnolol [22,23,24,25,26,27,28,29].

### 5.1. Anti-Inflammatory Effects of Carvacrol and Magnolol

It is known that cytokines are the link between cell damage and signs of inflammation (cell migration, edema, fever, or hyperalgesia) [107,108,109]. Cytokines are produced and released by many cell types in response to inflammatory stimuli. Inflammatory cytokines, such as IL-1β and TNF-α, are followed by the appearance of anti-inflammatory cytokines, IL-10, or interleukin-4 (IL-4) [109]. It is documented that cytokines such as TNF-α, IL-1β, and interleukin-17 (IL-17) play a substantial role in the inflammatory response [110]. TNF-α is secreted by mononuclear phagocytes and can induce acute phase proteins [111]. IL-17 produced by T helper 17 cells (Th17), natural killer (NK) cells, and neutrophil cells can exacerbate inflammation by proliferating the number of immune cells and indirect recruitment of neutrophils [112]. IL-1β is recognized as an acute phase mediator of the inflammatory response against infections [113].

Therefore, the ratio between pro- and anti-inflammatory cytokines modulates the intensity of inflammation [114]. Inflammation is also characterized by oversynthesis of inducible nitric oxide synthase (iNOS), oversynthesis of cyclooxygenase-2 (COX-2), and excess synthesis of nitric oxide (NO) and prostaglandins (PGE) [115]. Carvacrol has been shown to inhibit inflammatory cytokine levels and the expression of iNOS and COX-2 [116,117]. Other research has displayed that carvacrol inhibits neutrophil elastase production and the production of PGE2, prostaglandins F1 (PGF1), and prostaglandins F2 (PGF2) [114,117,118].

da Silva Lima et al. (2013) demonstrated in an in vivo animal study that the administration of carvacrol, in doses of 50–100 mg/kg, has an anti-inflammatory effect, attenuates inflammatory edema in rat paws, and reduces IL-1β and PGE2. At the same time, they demonstrated that the administration of a dose of 100 mg/kg reduces COX-2 and IL-1β messenger ribonucleic acid (mRNA) expression. Levels of IL-10 and anti-inflammatory cytokines were increased by carvacrol, which highlights the protective effect of this natural extract [114]. The anti-inflammatory effect of carvacrol may be due to the inhibition of one or both of the cyclooxygenase (COX) enzymes, an effect previously suggested in other studies, which shows the inhibitory effect of carvacrol on cyclooxygenase-1 (COX-1) and COX-2 [18,117]. Another study indicates that carvacrol plays an anti-inflammatory role by inhibiting inflammatory edema and leukocyte migration [119].

Tabibzadeh Dezfuli et al. (2017) also demonstrated that oral administration of carvacrol, once daily, in animals with streptozotocin (STZ)-induced diabetes, reduces the levels of IL-1β, IL-6m and TNF-α [120]. On the other hand, contradictory results were obtained, claiming that carvacrol has a positive effect in reducing IL-1β, IL-4, and IL-8, but would not have an effect on IL-6 and TNF-α, probably due to the methodology used in the studies by de Carvalho et al. (2020) [119,121].

In research on human subjects, Xiao et al. (2018) showed that carvacrol is able to inhibit the production of NO and PGE2, induced by IL-1β, but it also reduced the expression of iNOS, COX-2, and MMPs in chondrocytes by suppressing the signaling pathway NF-κB [122].

The anti-inflammatory characteristics of magnolol have also been investigated in abundant conditions. Magnolol exerts anti-inflammatory activity by inhibiting the formation of reactive oxygen species (ROS), COX-2 and iNOS expression, activating NF-κB, a transcription factor that directs inflammation in inflammatory diseases induced by LPS, and inhibiting the formation of pro-inflammatory cytokines [23,27].

In vitro studies coordinated by Lai et al. (2011) suggested that a dose of 5–15 μM magnolol may exhibit anti-inflammatory activity in LPS-induced RAW 264.7 cells. At the same time, magnolol inhibited iNOS and COX-2 gene and protein expression [123]. In another study, Lu et al. (2015) concluded that a dose of 5–20 μM magnolol significantly reduced inflammation, decreased the production of pro-inflammatory nitrates and PGE2, reduced iNOS and COX-2 expression, and activated NF-κB. At the same time, nuclear factor erythroid 2-related factor 2 (Nrf2) and hemogen oxygenase (HO) expression increased [124].

In an in vivo study by Lin el al., intraperitoneal (IP) injection of 20 mg/kg magnolol was shown to significantly improve the inflammatory response in Sprague–Dawley rats. Magnolol can attenuate ROS production, iNOS, and COX-2 expression, and NF-κB activation, as well as up-regulate of peroxisome proliferator-activated receptor gamma (PPAR-γ) expression [125]. Magnolol administered IP by Yang et al. (2016) has been shown to develop therapeutic potential in retinal angiogenesis and glial dysfunction, by decreasing inflammatory cytokines [126].

Research by Lu et al. (2013) on male rats of the Sprague–Dawley breed, with ligature-induced experimental periodontitis, showed that oral administration of *Magnolia officinalis* extract for 9 days inhibited neutrophil migration, myeloperoxidase (MPO) activity, COX-2 expression, and iNOS in gingival tissue [104]. In another investigation, Lee et al. (2005) highlighted the anti-inflammatory activity of magnolol and honokiol on a pathogenic anaerobic, *Propionibacterium acnes* (*P. acnes*), responsible for acne. This time, magnolol has been shown to inhibit NF-κB from COX-2, IL-8, and TNF-α promoters [127].

The abovementioned reveal that carvacrol and magnolol could be used successfully in the treatment of various inflammatory conditions, such as periodontal disease, whose main component is chronic inflammation, affecting the superficial and deep periodontal tissue, endangering the support of the tooth in the dental alveolus. Table 1 presents the anti-inflammatory effects of carvacrol and magnolol and Figure 3 shows the anti-inflammatory mechanism of carvacrol and magnolol in periodontitis and diabetes.

### 5.2. Antioxidant Properties of Carvacrol and Magnolol in Association with Periodontal Disease and Diabetes Mellitus

Carvacrol has strong antioxidant properties and can be effective in preventing and inhibiting many pathologies [128]. Oxidative stress is a substantial mechanism that may be involved in cytotoxicity induced by chronic stress [129].

Oxidative stress is the expression utilized for pathologies caused by ROS, imposed by free radicals. Oxidative stress is defined as the imbalance between oxidants and antioxidants, in favor of oxidants, with destructive and pathogenetic potential. Depending on the intensity, oxidative stress can occur intra or extracellularly. Intracellular oxidative stress can cause cell necrosis or more or less marked disorganization of the cell, with catastrophic effects in the case of a cell that cannot reproduce. Extracellular oxidative stress is cytotoxic, too. Free radicals are substances derived from incompletely oxidized compounds, which have undergone partial combustion, with oxygen groups in their structure capable of initiating aggressive oxidation reactions on the surface of cell membranes or even inside cells [130].

Rapid metabolism entrains additional free radicals, producing an imbalance between ROS generations and the antioxidant system. These free radical species lead to oxidative damage to various cells such as lipids, proteins, or nucleic acids [129]. The first-line defense antioxidants basically include superoxide dismutase (SOD), catalase (CAT), and glutathione peroxidase (GPx) [131].

Carvacrol treatment significantly improves glutathione (GSH) levels. The maintenance of GSH levels by carvacrol occurs mainly due to the removal of ROS, through its radical elimination effects [132]. Carvacrol has also been shown to increase antioxidant capacity in cell cultures and animals [133]. Oregano extract has been proved to have a protective effect against the function of free radicals, with the ability to prevent tissue damage, induced by chronic stress [9]. The harmful effects of chronic stress have been demonstrated to be ameliorated by carvacrol treatment. Carvacrol prevents lipid peroxidation by inducing SOD, GPx, glutathione reductase (GR), and CAT. Carvacrol effectively eliminates free radicals, such as peroxyl radicals, superoxide radicals, hydrogen peroxide, and NO [134,135].

Carvacrol exerts antioxidant effects in vitro and in vivo, and the antioxidant activity is attributed to the presence of the –OH, related to the aromatic ring [136,137]. Another study, conducted by Samarghandian et al. (2016), showed that carvacrol inhibits oxidative damage to the brain, liver, and kidneys, being a new pharmacological agent, fruitful for relieving oxidative damage, induced by chronic stress [9].

In research by Tabibzadeh Dezfuli et al. (2017) it was shown that oral administration of 15 mg/kg body carvacrol, per day, to diabetic rats, can lower malondialdehyde (MDA) levels and increase CAT, SOD, and GPx activity, compared to rats to whom the extract has not been administrated, suggesting that carvacrol has antioxidant properties [120].

It is verified that the accrual of oxidative stress plays a critical role in the aggravation of both pathologies: Periodontitis and diabetes. In addition to its anti-inflammatory effects, magnolol has also antioxidant properties. Besides, it has been stated that magnolol scavenges hydroxyl radical [136], peroxy-nitrite [138], and hydrogen peroxide [139] to reduce or suppress the generation of ROS. As well, there is additional direct proof of its anti-oxidative effect on intracellular GSH depletion [25] or enzymatic system capacity in the rat model [140].

A study conducted by Zhao et al. (2016) shows that magnolol supplements are useful in wound healing and modulating inflammation by decreasing Nrf2 in patients with diabetes and periodontitis. Nrf2 is a transcription factor with a crucial role in regulating the antioxidant response and has been decreased in oral neutrophils in patients with aggressive periodontitis [141].

Nrf2 and hemogen oxygenase-1 (HO-1), one of its main target genes, have been proved to be in charge of the increased inflammation in osteoarthritis associated with NIDDM [142]. Several studies have suggested that Nrf2 is crucial in regulating antioxidants in patients with advanced periodontitis [143]. Therefore, treatments targeting the Nrf2/HO-1 axis can relieve oxidative stress and inflammation in diabetic patients with periodontal disease [144].

Other authors have shown that magnolol amplifies the expression of the Nrf2/HO-1 axis, depending on the dose, suggesting that magnolol can attenuate ROS generation by activating Nrf2/HO-1 signaling. They also proved that magnolol reduces the production of two cytokines, IL-6 and IL-8. The production of IL-6 and IL-8, induced by AGEs, has been prevented with Nrf2, concluding that the increase in Nrf2 suppresses these cytokines [144]. On the other hand, the production of ROS induced by AGEs also decreased after the administration of the magnolia extract. Furthermore, magnolol has been illustrated to activate Nrf-2/HO-1 signaling and suppress inflammation induced by *P. gingivalis* LPS in macrophages [145].

Proteins and lipids are often subjected to irreversible non-enzymatic glycosylation in patients with chronic hyperglycemia, leading to the formation of AGEs. The role of AGEs in potentiating diabetic complications is discussed by activating cellular responses by AGEs-modified proteins, which interact with specific receptors on the cell surface [146]. A recent study shows that in patients with marginal periodontitis and diabetes, the levels of AGEs in the blood are significantly increased [147].

In in vitro examinations, the use of a dose of 16 μM magnolol reduced the oxidative stress caused by acrolein in human SH-SY5Y cells and acted on the following signaling pathways: JNK (c-Jun N-terminal kinase), mitochondria, caspase, phosphoinositide 3-kinase (PI3K), mitogen-activated protein kinase (MEK), extracellular signal-regulated kinases (ERK), protein kinase B (Akt), and forkhead box protein O1 (FOXO1). It also inhibited the accumulation of ROS and the accumulation of intracellular GSH [148]. It has been found that IV administration of 20 mg/g magnolol could significantly reduce MPO activity and the expression of TNF-α, IL-6, and iNOS, so as to inhibit oxidative stress [29].

Consequently, magnolol down-regulates MPO activity, TNF-α, IL-6, and iNOS expression due to JNK, mitochondria, caspase modification, and PI3K, MEK, ERK, Akt, and FOXO1 signaling pathways [29,148].

### 5.3. Antimicrobial Activity of Carvacrol and Magnolol against Periodontal Pathogens

Carvacrol, similar to thymol, acts on microbial cells and causes structural and functional damage to bacterial membranes [11]. Carvacrol is one of the few elements of essential oil that has the ability to dissolve the outer membrane of gram-negative bacteria, causing the release of LPS [149,150].

Research conducted by Wang and et al. (2016) evaluated the antibacterial character of the phenolic components of oregano essential oil against oral microorganisms. The components evaluated were hinokitiol, carvacrol, thymol, and menthol. In this study, the minimum inhibitory concentration (MIC) and minimum bactericidal concentration (MBC) of these components were demonstrated, as a result of the fact that carvacrol has an MIC of 200–400 µg/mL and an MBC of 200–600 µg/mL on *A. actinomycetemcomitans*, *Streptococcus Mutans* (*S. Mutans*), *Methicillin-resistant Staphylococcus aureus* (MRSA), and *Escherichia. Coli (E. Coli*) [151].

Another analysis, conducted by Maquera-Huacho et al. (2018) evaluated the antibacterial properties of carvacrol and terpinen-4-ol against *P. gingivalis* and *F. nucleatum* and the cytotoxic effect on fibroblasts. The MIC and MBC of carvacrol were 0.007% for *P. gingivalis* and 0.002% for *F. nucleatum*. The results showed anti-biofilm activity of carvacrol (0.26%, 0.06%) and the cytotoxicity was similar to that of chlorhexidine (CHX).

Therefore, the authors demonstrated that carvacrol has antibacterial activity on the periodontal biofilm [152].

In an in vitro study, Lu et al. (2013) showed that *Magnolia officinalis* extract inhibited key pathogens in the initiation of periodontal disease, *P. gingivalis* and *A. actinomycetemcomitans* [104].

Ho et al. (2001) demonstrated through in vitro studies the MIC of magnolol in order to exert the antimicrobial effect. At an MIC dose of 25 µg/mL, magnolol has a marked antimicrobial effect against *P. gingivalis*, *A. actinomycetemcomitans*, *Prevotella intermedia* (*P. intermedia*), *Micrococcus luteus* (*M. luteus*), and *Bacillus subtilis* (*B. subtilis*), so it can be used as an adjunct in the treatment of periodontitis [145].

Another in vitro study determined the MIC and MBC of honokiol and magnolol on oral bacteria. Thus, Chiu et al. (2021) discovered that the MIC of magnolol was 10 µg/mL for *A. actinomycetemcomitans* and the MBC of magnolol was 20 µg/mL, 20 µg/mL, and 30 µg/mL for *A. actinomycetemcomitans*, *S. mutants,* and MRSA [153].

The results of the studies presented above emphasize the antimicrobial properties of carvacrol and magnolol on periodontal pathogens. The MIC and MBC of carvacrol and magnolol for the periodontal pathogens are summarized in Table 2 and Table 3.

### 5.4. Anti-Osteoclastic Properties of Carvacrol and Magnolol

One of the key mediators of bone resorption, of a local or general nature, is chronic inflammation. Chronic inflammation is found in both periodontitis and diabetes; therefore, these patients have an increased risk of osteoporotic fracture [154].

Bone tissue is a dynamic tissue that undergoes continuous remodeling, generated by the activity of osteoblasts and osteoclasts. Osteoclast activity is increased in periodontitis, with adverse consequences on bone trabeculae and alveolar ridges. Osteoclasts appear as an inflammatory response to the production of receptor activators of nuclear factor-kappa B ligand (RANKL) and bacterial LPS, generated by cytokines [14].

One of the inflammatory biomarkers is IL, whose values are increased in pathologies such as periodontitis [155]. IL-1 directly stimulates bone resorption through osteoclasts, while prolonging their lifespan [156,157]. Osteoclasts are multinucleated cells, differentiated from monocytes or macrophages, involved in bone resorption [158]. IL-1 has an indirect effect on osteoclast differentiation by increasing RANKL expression and decreasing osteoprotegerin (OPG), an inhibitory factor in osteoclastogenesis. OPG is a trap receptor for RANKL in osteoblasts [159].

Carvacrol abolished the RANKL-induced formation of tartrate-resistant acid phosphatase (TRAP)-positive multinucleated cells in RAW 264.7 macrophages and human CD14+ monocytes. Moreover, oregano extract inhibited LPS-induced osteoclast formation in RAW 264.7 macrophages. Exploration of the underlying molecular mechanisms revealed that carvacrol down-regulated RANKL-induced NF-κB activation in a dose-dependent manner. Furthermore, the suppression of NF-κB activation is correlated with the inhibition of inhibitor of kappa B kinase (IκB) activation and the attenuation of inhibitor of kappa B kinase α (IκBa) degradation. Carvacrol potentiated apoptosis in mature osteoclasts by caspase-3 activation and DNA fragmentation. Furthermore, carvacrol did not influence the viability of proliferating MC3T3-E1 osteoblast-like cells. Together, these results founded by Deepak et al. (2016) demonstrate that carvacrol mitigates osteoclastogenesis by damaging the NF-κB pathway and induction of apoptosis in mature osteoclasts [14].

A study by Bothelo et al. (2009) showed that local treatment with carvacrol gel significantly inhibited bone resorption in the alveolar bone of rats with experimentally induced periodontal disease. The topical application of carvacrol gel also inhibited the multiplication of periodontal microorganisms in these animal models [102].

Magnolia extract has been shown to modulate the resorption of alveolar bone in rats. In the same study, it was revealed that magnolol reduces RANKL expression, reduces gingival inflammation, and decreases the number of osteoclasts [104]. A dose of 5–20 µM magnolol has been shown to be effective in inhibiting RANKL-induced osteoclast differentiation from macrophage-like cells called RAW 264.7 cells [104,124]. On the other hand, a dose of up to 20 µM does not influence the differentiation of osteoclasts induced by RANKL expression in rats [160].

In vitro cell culture research has shown that magnolol inhibits IL-1 induced osteoclast differentiation, decreases RANKL expression in IL-1 stimulated osteoblasts, and reduces IL-1 induced PGE2 production, by inhibiting COX-2 expression. Magnolol restrains the formation of osteoclasts [161].

Hwang et al. (2018) studied the effect of magnolol on osteoclast differentiation and concluded that *Magnolia officinalis* extract prevents the formation of IL-1 induced osteoclasts by the following mechanisms: Inhibition of COX-2 expression, inhibition of PGE2 synthesis, and suppression of RANKL expression. PGE2 mediates bone resorption generated by inflammation and *Magnolia officinalis* extract inhibits PGE2 synthesis, thus demonstrating the protective effect of magnolol on marginal periodontitis [161]. In addition to the anti-osteoclastogenetic effect, magnolol stimulates the differentiation of osteoblasts and their proliferation [22].

A dose of 0.1 μM magnolol modulated the production of factors that induce osteoclast differentiation, such as TNF-α, IL-6, and RANKL [22]. In RANKL-induced RAW 264.7 cells, 75–150 μM magnolol reduces osteoclast differentiation, TRAP activity of differentiated cells, and the area of bone resorption generated by osteoclasts [104]. Furthermore, 2.5–20 μM magnolol attenuates RANKL-induced osteoclast differentiation by inhibiting ROS production, suppressing mitogen-activated protein kinase (MAPK), C-proto-oncogene (c-fos), activator protein-1 (AP-1) and NF-κB, and increasing HO-1 expression [124].

In rats in which periodontal disease was experimentally induced by ligature, the administration of 100 mg/kg magnolol per os (PO) significantly decreased the resorption of the alveolar bone, the volume of osteoclast cells at the level of the alveolar ridge, as well as the RANKL expression. The same dose of magnolol could reduce the expression of MMP-9, MMP-1, iNOS, and the activation of TNF-α and COX-2 [104]. At the same time, in the case of experimental periodontitis induced in rats, an increase in neutrophil infiltrate was observed. Following the administration of magnolol extract, the values of superoxide, NF-κB, iNOS, COX-2, MMP-1, and MMP-9 in the gingival tissues decreased. Magnolol significantly reduced the resorption of the alveolar ridge in rats with experimentally induced periodontitis, suppressing the accumulation of periodontal bacteria, suppressing the synthesis of the inflammation mediator mediated by NF-κB, reducing RANKL, and therefore partially blocking osteoclast formation [104].

Thus, magnolol has a number of activities to stimulate osteoblasts and inhibit osteoclasts in cell cultures and has been recommended for screening for anti-osteoporosis activity [104].

### 5.5. Anti-Diabetic Properties of Carvacrol and Magnolol

There is little knowledge about the effects of carvacrol on diabetes. However, Bayramoglu et al. (2014) evaluated the anti-diabetic properties of carvacrol in STZ-induced diabetic rats. Following PO administration of carvacrol doses of 25 mg/kg body weight (BW) and 50 mg/kg BW, there was a reduction in serum glucose, a significant reduction in total plasma cholesterol (TC), and a reduction in aspartate aminotransferase (AST), alanine aminotransferase (ALT), and lactate dehydrogenase (LDH). Therefore, it has been established that this extract provides partial protection against liver enzymes [162].

In another experimental study, Li et al. (2020) used lower doses, 10 mg/kg BW and 20 mg/kg BW administered IP in mice and after 4 to 6 weeks administration determined the following values: TC, triglycerides (TG), AST, ALT, alkaline phosphatase (ALP), LDH, and the activity of liver enzymes involved in glucose metabolism. They showed that regardless of the dose administered, carvacrol decreased blood glucose levels and the 20 mg/kg dose significantly reduced LDH plasma levels. They concluded that this extract exerts anti-hyperglycemic effects in rats with experimentally induced diabetes [163].

Another study showed that the combination of rosiglitazone (RSG), an antidiabetic drug of the thiazolidione family, and carvacrol has antihyperglycemic effects, beneficial in improving carbohydrate metabolism, in mice with NIDDM who were given a diet rich in fats [164].

Several studies have shown that major bioactive constituents of *Magnolia officinalis* extract contribute to glycemic control [165,166]. Studies in diabetic rats have proved that magnolol is effective against oxidative damage to the liver, modulating hyperglycemia and hyperlipidemia. At the same time, it has been stated to inhibit the activity of cytochrome (P450 2E1), a mechanism against insulin resistance [167].

Wang et al. (2014) studied the effects of magnolol on hyperglycemia, hyperlipidemia, and hepatic oxidative stress in a diabetic model in rats, established using STZ and a high-fat diet (HFD). Following PO administration of doses of 25, 50, and 100 mg/kg for 10 weeks, the values of TC, TG, and low-density lipoprotein (LDL) cholesterol decreased significantly, while the antioxidant liver enzymes (CAT, GSH) increased. These results illustrate that magnolol is effective against liver damage induced by oxidative stress, acting as support against hyperglycemia and hyperlipidemia [167]. Simultaneously, in other in vivo studies, oral treatment with 200 mg/kg honokiol, the isomer of magnolol, for 8 weeks significantly decreases fasting blood glucose in NIDDM mice [168].

Most studies have shown that the anti-inflammatory and antioxidant effects of magnolol are closely correlated with the preventive effects against diabetes and its complications [141]. The anti-diabetic properties of carvacrol and magnolol in in vivo animal model research are shown in Table 4.

### 5.6. Toxicity of Carvacrol and Magnolol

Most drugs have the ability to develop potential side effects, and natural extracts do not make an exception. It is difficult to record the toxicity of essential oils, as the toxicity varies depending on the number of components that compose the aromatic oil [169]. However, essential oils are considered safe for consumption, as a natural substitute for antioxidant food additives [170].

Kohlert et al. (2002) have shown that the most toxic concentration of carvacrol is 36–49 mg/L [171]. Carvacrol at a concentration of up to 25 μM in V79 fibroblast cells in the lungs of hamsters did not cause DNA damage, according to measurements made by Undeger et al. (2009) [172].

Suntres et al. (2015, 2020) identified the average lethal doses of carvacrol in rats: A dose of 810 mg/kg in oral administration, 80 mg/kg in IV administration, and 73 mg/kg in IP injection [10,173]. In 2017, Kuo et al. (2017) administered IG with a dose of 70 mg/kg of carvacrol in rats and found that they suffered from dehydration, diarrhea, or even mortality, 2–3 days after treatment, which emphasize carvacrol toxicity at this dose [101].

Caco-2 intestinal cells were used to study the mutagenic and genotoxic effect of carvacrol. Elevated doses of 460 μM carvacrol were used and these caused damage to the purine bases in DNA [174,175] but had no adverse effects on hamster lung fibroblasts or human hepatocytes and lymphocytes [174,175,176].

In their studies regarding magnolol toxicity, conducted in 2006, Saito et al. (2006) found that magnolol extract did not show mutagenic toxicity and genotoxicity [177]. A more recent study, conducted by Sarrica et al. (2018), showed through in vivo and in vitro experiments that concentrated magnolia root extract (MRE) has no mutagenic or genotoxic potential, while an OECD (Organisation for Economic Co-operative and Development) study established that no adverse effects occur at MBE (magnolia bark extract) concentrations >240 mg/kg, thus being considered safe for consumption [178].

Studies in humans have shown that dietary supplementation with magnolol affects only 1/22 patients, with symptoms such as heartburn, thyroid dysfunction, or shaking hands, but the link between these symptoms and the treatment could not be explained [179]. Another experiment by Mucci et al. (2006) included 89 postmenopausal women who received 60 mg of MRE and 50 mg of magnesium. The treatment was tolerated by 94% of subjects without side effects [180]. However, studies by Teschke et al. (2014–2016) have reported that some *Magnolia*-based mixtures may be hepatotoxic [181,182,183].

Carvacrol has been approved for food use by the Food and Drug Administration. It has been included by the Council of Europe in the list of approved chemical flavorings [184,185]. This extract is also used in food, spice, or pharmaceutical industries [170]. Nevertheless, different institutions have comprised *Magnolia officinalis* in lists of herbal preparations suitable for inclusion in food supplements, because of its digestive and rebalancing activity upon the oral microbiome. In the market, anti-ageing cosmetics containing magnolol have also appeared [186].

Therefore, when the doses are obeyed, the two natural extracts, carvacrol and magnolol, can be considered safe, but further research is needed to determine their toxicity when administered in periodontitis and diabetes.

## 6. Conclusions and Prospective

Our manuscript reviewed the results of various surveys on the topic and emphasized the therapeutic effects of carvacrol and magnolol on periodontal disease and diabetes mellitus. Following this analysis, it is obvious that carvacrol and magnolol have beneficial properties in the investigated pathologies, as demonstrated by in vitro and in vivo studies. These natural extracts have potential as a future “key-role player” that can be integrated into new treatment formulas, effective both in reducing gingival inflammation and periodontal pockets, while also controlling blood sugar in diabetic patients. For example, a new treatment perspective could be the development of a periodontal gel containing magnolol as an active ingredient, as the literature has only revealed the use of a topical carvacrol-based periodontal gel.

We believe that this literature review is of interest because no papers have been published that evaluate the potential benefits of both extracts in the same study. However, future studies are needed to maximize the therapeutic potential of carvacrol and magnolol to bring these compounds to the clinic.

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
