# Peer review of "Anti-Inflammatory and Antioxidant Properties of Carvacrol and Magnolol, in Periodontal Disease and Diabetes Mellitus"

_molecules, 2021, doi:10.3390/molecules26226899_

Round 1

Reviewer 1 Report

Honokiol often shows higher biological activity than magnolol. Please explain why you are describing only magnolol and not honokiol for lignans in this review.

Author Response

Honokiol often shows higher biological activity than magnolol. Please explain why you are describing only magnolol and not honokiol for lignans in this review.

Response: The properties of magnolol were the focus of our review as we intend to test this extract in vivo, in rats with experimentally induced periodontal disease. We investigated it’s effects on periodontal disease and diabetes mellitus. We chose this extract for further in vivo studies due to it's antibacterial activity against P. gingivalis and A. actinomycetemcomitans, the protective effect on bone resorption in the alveolar bone crest demonstrated by micro CT investigations and the anti-inflammatory activity by inhibiting neutrophil migration, MPO activity and COX-2 and iNOS expression in gingival tissue.

Reviewer 2 Report

The presented article is endowed with important practical value. A thorough analysis of the data provided in the article was carried out. At the same time, I would like to propose to change the title of the article to emphasize the anti-inflammatory properties of bioactive plant extracts, carvacrol and magnolol, in periodontal diseases following diabetes mellitus. Especially, it is advised to highlight clinical studies (if they were conducted) and pay special attention to the evidence of the above pharmacological properties of carvacrol and magnolol (Fig.4). It is necessary to draw the attention of the authors that plant extracts are by definition natural and it is redundant to mention them as "natural plant extracts". Therapeutic properties should only be assessed when studying the long-term effect of compounds according to standard schemes. It is necessary to expand the information on the pharmaceutical formulations of the extracts used.

Author Response

The presented article is endowed with important practical value. A thorough analysis of the data provided in the article was carried out. At the same time, I would like to propose to change the title of the article to emphasize the anti-inflammatory properties of bioactive plant extracts, carvacrol and magnolol, in periodontal diseases following diabetes mellitus.

Response 1: Thank you for your kind suggestion. We modified the title according to the recommendations.

Especially, it is advised to highlight clinical studies (if they were conducted) and pay special attention to the evidence of the above pharmacological properties of carvacrol and magnolol (Fig.4).

Response 2: We kindly appreciate your suggestion. Please check paragraph 5.1-5.5 where we highlighted the pharmacological properties of carvacrol and magnolol.

It is necessary to draw the attention of the authors that plant extracts are by definition natural and it is redundant to mention them as "natural plant extracts".

Response 3: All references to “natural plant extracts” have been removed from the text.

Therapeutic properties should only be assessed when studying the long-term effect of compounds according to standard schemes. It is necessary to expand the information on the pharmaceutical formulations of the extracts used.

Response 4: Thank you for this suggestion. We summarized the treatment doses and the protocol of administration in table 1. 

Reviewer 3 Report

The article entitled " Therapeutic Properties of Bioactive Natural Plant Extracts, Carvacrol and Magnolol, Against Periodontal disease and Diabetes mellitus" written by Georgiana Ioana Potra Cicalău et al., is found interesting. Georgiana Ioana Potra Cicalău et al., explored the therapeutic potential of Carvacrol and Magnolol against the periodontal disease and Diabetes mellitus.

The article has been written very nicely, however needs major revision 

  • Carvacrol is considered as an essential oil of aromatic plants such as Origanum vulgare, therefore writing plant extract in the title should be reconsider.
  • Line No-22-23; kindly recheck this sentence and put full stop.
  • A flowchart should be added to the article to show the research methodology. Also include a section on the bibliometric information to understand the repeatability of the study. Authors could take help of https://doi.org/10.1016/j.scitotenv.2021.144990.
  • Text presented in Figure 3 can be made bold or the color chosen must be in such a way so the content should be clearly visible.
  • Line No-277-279 scientific name presented should be uniform throughout the manuscript. Either use italic font or regular
  • Line no-362; 373, 375 da Silva Lima et al. (………….), Tabibzadeh Dezfuli et. al., Kindly add year along with author names throughout the manuscript or check the journal guidelines
  • Add all the references (Line no-378)
  • It is recommended to add a mechanistic figure to understand the anti-inflammatory approach of carvacrol and magnolol in both the diseased condition.
  • Table 1 legend should be placed at appropriate place.
  • In Table 1, one more column can be added indicating separately doses of treatment used for carvacrol and magnolol instead of writing in column two.
  • Subheading 5.2: Antioxidant properties of carvacrol and magnolol presented should more specific in association with Periodontal disease and Diabetes mellitus
  • Line no-537; Kindly check the serial No of table and table legends should be placed as per journal guidelines
  • Line no-535; Table 2 is missing in the manuscript, kindly provide the details in the revised manuscript
  • Table 4; one more column can be added indicating separately doses of treatment used for carvacrol and magnolol instead of writing in column two.
  • In Table 4 & Table 1; kindly years of study along with author details
  • Line No 676: dietary supplementation with Magnolia officinalis extract affects- Does this extract reflect about the magnolol?
  • In general: authors are recommended to choose more scientific words rather than general vague term to describe any important aspects.
  • Over all, English language should be thoroughly checked, at several places rephrasing of sentences required. In addition unnecessary commas should be removed from the sentences.

Round 2

Reviewer 1 Report

This review has been well improved.

Reviewer 3 Report

The article entitled " Anti-Inflammatory and Antioxidant Properties of Carvacrol and Magnolol, in Periodontal disease and Diabetes mellitus " written by Georgiana Ioana Potra Cicalău et al., is found interesting. Georgiana Ioana Potra Cicalău et al., explored the therapeutic potential of Carvacrol and Magnolol against the periodontal disease and Diabetes mellitus. Authors have revised the manuscript and now it looks fine. Therefore, I would like to recommend this manuscript for publication